# Organic Contaminants in Zooplankton of Italian Subalpine Lakes: Patterns of Distribution and Seasonal Variations

Simona Pascariello [1], Michela Mazzoni [1,2,*], Roberta Bettinetti [2] , Marina Manca [3],
Martina Patelli [4], Roberta Piscia [3], Sara Valsecchi [1] and Stefano Polesello [1,*]

[1]  Water Research Institute (IRSA)—CNR, Via del Mulino 19, 20861 Brugherio (MB), Italy;
   pascariello@irsa.cnr.it (S.P.); valsecchi@irsa.cnr.it (S.V.)
[2]  Department of human sciences and innovation for the territory (DISUIT), University of Insubria, Via S.
   Abbondio 12, 22100 Como (CO), Italy; roberta.bettinetti@uninsubria.it
[3]  Water Research Institute (IRSA)—CNR, Largo Tonolli 50, 28922 Verbania Pallanza (VB), Italy;
   marina.manca@irsa.cnr.it (M.M.); roberta.piscia@irsa.cnr.it (R.P.)
[4]  Department of earth and environmental sciences (DISAT), University of Milano-Bicocca, Piazza della
   Scienza 1, 20126 Milano (MI), Italy; m.patelli3@campus.unimib.it
*  Correspondence: mmazzoni@studenti.uninsubria.it (M.M.); polesello@irsa.cnr.it (S.P.)

**Abstract:** Zooplankton is a key node in many trophic webs, both for food that for persistent organic contaminants that can accumulate in biota. Zooplankton of different size was seasonally sampled for two years in three deep Italian subalpine lakes (Maggiore, Como, Iseo) with the aim of determining the concentrations of perfluoroalkyl substances (PFAS), DDT, and PCB, and assessing the seasonality impacts on contaminants concentrations. In general, Lake Maggiore showed the highest concentrations for each group of contaminants, with mean values of 7.6 ng g$^{-1}$ ww for PFAS, 65.0 ng g$^{-1}$ dw for DDT, and 65.5 ng g$^{-1}$ dw for PCB. When considering the composition pattern, perfluorooctane sulfonate (PFOS) was detected in 96% of the samples and it was the predominant PFAS compound in all of the lakes. pp' DDE was the most detected congener among DDTs and their metabolites, while for PCBs, the prevalent group was hexa-CB that constituted 35.4% of the total PCB contamination. A seasonal trend was highlighted for all contaminant groups with concentrations in colder months greater than in spring and summer; it was evident that the contaminant concentrations were more dependent from seasonality than from size, trophic levels, and taxa composition of zooplankton. Principal component analysis showed that one of the main driver for the accumulation of most of the studied contaminants is their lipophilicity, except for perfluorooctanoic acid (PFOA) and octachlorobiphenyl.

**Keywords:** zooplankton; perfluoroalkyl substances (PFAS); organochlorine compounds; lakes

## 1. Introduction

In aquatic food webs, zooplankton has an important role, because it transfers energy and organic matter from basal producers (phytoplankton and bacteria) to higher trophic levels up to large predators [1]. In the trophic chain, zooplankton is also a source of contaminant exposure for predators, but its role in the processes of bioaccumulation/biomagnification is not well elucidated. Moreover, in the ecotoxicological model, zooplankton was usually considered as a single functioning entity, despite the richness of taxon, sizes, and trophic levels that compose this heterogeneous group. Only recently some zooplankton subsets have been the subject of specific studies [2], which showed that size fractions (e.g., mesozooplankton and macrozooplankton) differ in their taxonomic, elemental,

and biochemical composition [3], and their contaminant concentrations could vary with the size, as described for MeHg [4]. Size is not the only important variable, but it is also necessary to consider the abundance, biomass, and composition of zooplankton community for a significant evaluation of contamination data [5]. For example, Taylor et al. [6] found a negative relationship between plankton biomass and DDT and PCB concentrations in Ontario Lake, which is stronger for more hydrophobic compounds. Back et al. [7] described the decrease of mercury levels (about 50–70%) during spring and summer, concurrently with the biomass increase; however, they could not discriminate whether the mercury concentrations were diluted by the increase of zooplankton or phytoplankton biomasses.

In the large sub-alpine (or perialpine) lakes, zooplankton was widely investigated, mainly focusing on the food-webs characterization and the impact of eutrophication, oligotrophication, and climatic fluctuations [8]. Previous studies regarding chemical contamination were predominantly focused on polycyclic aromatic hydrocarbons (PAHs) and legacy compounds (e.g., DDT, PCB), especially in Lake Maggiore where zooplankton has been seasonally monitored since 2008 to track the variation in DDT and PCB levels [9].

DDT and PCB are organochlorine compounds that are banned in many countries since the 1970s and 1980s. These contaminants have been demonstrated to be persistent and bioaccumulative in the trophic chain for their chemical properties [10]. In fact, they are still widely detected in the water environment [11], like perfluoroalkyl substances (PFAS) [12]. These substances are synthetic chemicals that are utilised in many industrial and consumer products [13]; some of them (perfluorooctane sulfonate, PFOS, and related compounds) have been regulated in Annex B list of the Stockholm Convention on Persistent Organic Pollutants in 2009. The effects of PFOS and perfluorooctanoic acid (PFOA) on the structure of zooplanktonic community have been studied in laboratory microcosm [14,15], but few studies are devoted to its role in the trophic transfer of PFAS in freshwater food webs [16,17].

In the present study, zooplankton samples of different size were seasonally collected for two years in three deep subalpine lakes (Maggiore, Como, Iseo), with the aim to (i) determine the concentrations of DDT, PCB, and PFAS in the zooplankton of these subalpine lakes; (ii) assess the influence of some parameters of zooplankton community on contaminants concentrations (e.g., size fractions, biomass, feeding behaviour); and, (iii) identify external variables (e.g., contaminant sources, seasonality, temperature) that could influence the zooplankton bioaccumulation in lakes.

## 2. Material and Methods

### 2.1. Study Area

The deep lakes Maggiore, Como, Iseo are located within the River Po basin in the pre-alpine area (Figure 1) and constitute a large part of all freshwater Italian resources. Table S1 reports the main characteristics of the lakes. They have similar morphological features since they have the same fluvio-glacial origin. They are classified as olo-oligomictic, because they have long period of incomplete mixing during the spring and only occasional overturns after frosty and windy winters. Furthermore, the complete homogenization of their water has recently become rare and irregular because of climate change [18].

These lakes have a remarkable environmental value and satisfy the drinking water need of towns (e.g., Como and Lecco) and villages in the provinces of Como, Lecco and Brescia, as well as the agricultural and industrial water requests in large areas of Northern Italy. Furthermore, they sustain significant local economic activities, such as tourism and fishery.

Lake Como, the deepest Italian lake, is characterized by an "upside-down Y" shape (Figure 1); in the southern part a bathymetric ridge separates two branches: the deep western branch, with no outflow and a longer water renewal time; and, the more open eastern branch with an emissary (river Adda) and more regular bathymetry. Lake Iseo is the fourth largest Italian lake that is fed by waters coming from the Valcamonica Valley. The shoreline area is due to undergoing sewage treatment by two treatment plants that are located at the northern and southern ends of lake [19].

Lake Maggiore, the second largest and deepest Italian lake, is divided between Italy (Piedmont and Lombardy Regions) and Switzerland (Canton Ticino). Most of the population and the main industrial activities are in the southern part [20]. Until the 1990s, a chemical factory producing technical DDT and using a mercury-cell chloralkali plant discharged wastewaters into the Toce River, which carried pollutants to the lake, where DDT and Hg accumulated in sediment and biota, causing an important contamination [21].

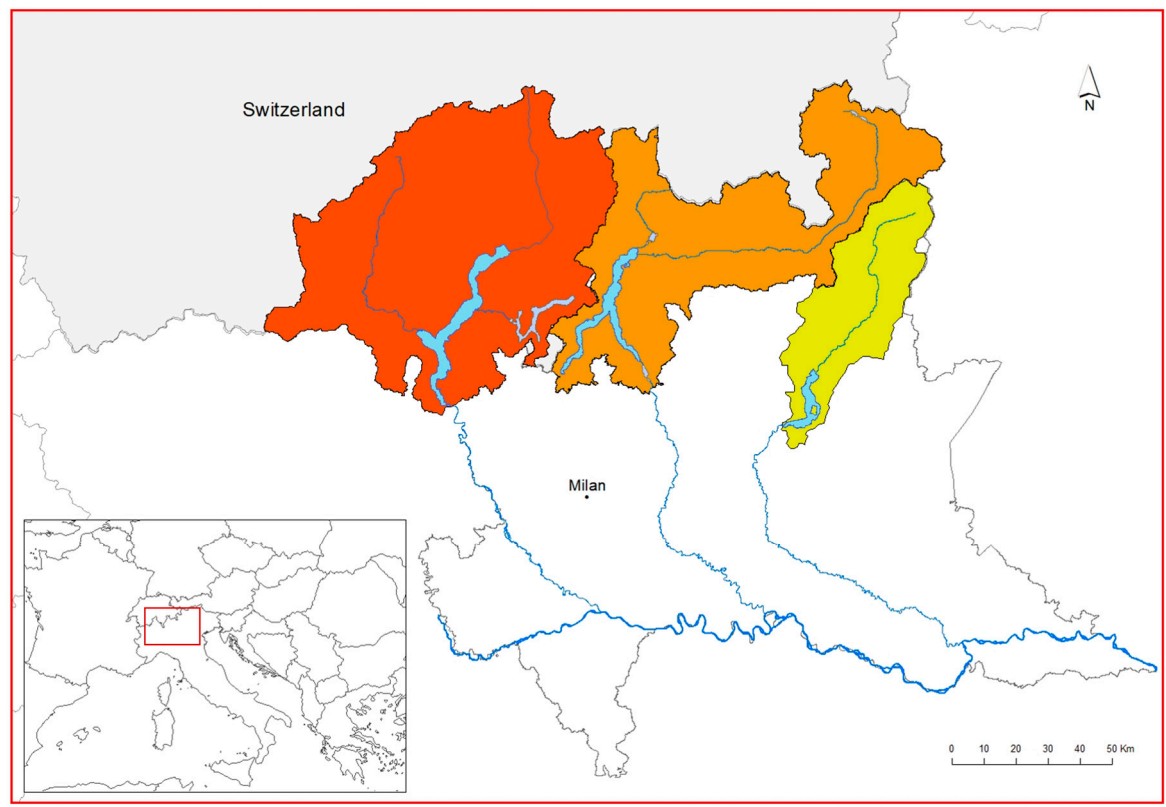

**Figure 1.** Study area. From west to east: Lake Maggiore (red area), Lake Como (orange), and Lake Iseo (yellow) basins. The map is obtained by ArcGIS® software.

*2.2. Sampling and Determination of Biomass*

Crustacean pelagic zooplankton was seasonally collected for two consecutive years (2016–2018) and they were analysed together with some samples that were archived from previous years, as described in Table S2. The samples were vertically caught from a boat in the middle of the lake using nylon nets with mesh of 200, 450, and 850 μm at 20 m depth. Those sizes were chosen in order to avoid large phytoplankton colonies and rotifer taxa, while the depth was chosen while considering the average transparency of lakes and where phytoplankton lives in order to collect most of the crustacean zooplankton community. Every sampling was repeated until we got sufficient biomass for analysis.

The collected zooplankton included Copepoda (Cyclopoida and Calanoida) and Cladocera (*Daphnia longispina* group, *Eubosmina coregoni*, *Diaphanosoma brachyurum*, *Leptodora kindtii,* and *Bythotrephes longimanus*).

Density and biomass were calculated using optical microscope at 40× and equations of length-weight regression for samples of Lake Como and Lake Maggiore [22].

For chemical analysis two aliquots of each sample were filtered on 2 μm pore glass-fibre filters (GF/C, 4.7 cm of diameter, Whatman, Maidstone, UK) and then frozen at −20 °C.

### 2.3. Chemical Analysis

### 2.3.1. Perfluoroalkyl Substances

For the analysis of zooplankton, about 5 g of wet weight sample was weighed and then spiked with 100 µL of 40 µg $L^{-1}$ stable isotope-labelled solution used as internal standard. SIL-IS was prepared from a mixture of mass-labelled MPFAC-MXA and mass-labelled M3PFPeA solutions purchased from Wellington Laboratories Inc. (Guelph, ON, Canada). The extraction method for PFAS is described in detail in [19]. Briefly, a mixture of water and acetonitrile (10:90 *v/v*) and few µL of formic acid were added to the spiked samples. The samples were subjected to ultra-sonication extraction, centrifugation, and a treatment with $MgSO_4$/NaCl. Afterwards, the extracts were partially evaporated and then filtered by HybridSPE®-Phospholipid Ultra cartridges (Merck KGaA, Darmstadt, Germany) to eliminate phospholipids. The final extracts were analysed by liquid chromatography coupled to mass spectrometry (UHPLC-MS/MS) after an online purification with turbulent flow chromatography (TFC).

External standard solutions at different concentrations were prepared by diluting PFAC-24PAR Standard Solution (Wellington Laboratories Inc., Guelph, ON, Canada) containing certified native PFAS in acetonitrile to obtain the calibration curve. The obtained solutions were acidified with 50 µL of concentrated formic acid and then spiked with 100 µL of SIL-IS. Limit of detection (LOD) and limits of quantification (LOQ) were estimated, according to the ISO Standard 6107-2:2006, as respectively, threefold and tenfold the standard deviation of an extract of biological tissue fortified at 1 µg $L^{-1}$, as described in [23].

Details on the analyte names, abbreviations, and corresponding SIL-IS are reported in Table S3. A full list of chemicals and solvent is provided in the Supporting Information.

### 2.3.2. Organochlorine Compounds

Organochlorine compounds (OC) were analysed following the method that is described in [21]. Briefly, each sample of zooplankton was freeze-dried, about 0.5 g were put into a glass fibre thimble (19 mm I.D., 90 mm length, Whatman, Maidstone, UK) and then extracted in a modified Soxhlet equipment (ECO 6 Thermoreactor, Velp Scientifica, Usmate, Italy) for two hours with a n-hexane and acetone (1:1) mixture (pesticide analysis grade, Carlo Erba Reagents s.r.l, Cornaredo, Italy). The lipid content was gravimetrically determined, and the extract was then digested with 2 mL of $H_2SO_4$ (98%, Carlo Erba Reagents s.r.l, Cornaredo, Italy) all night long. The supernatant was cleaned up on a Florisil® column (40 × 7 mm I.D.), eluted by 25 mL of a 85:15 mixture of n-hexane and dichloromethane (pesticide analysis grade, Carlo Erba Reagents s.r.l, Cornaredo, Italy) and, finally, concentrated to 0.5 mL. The analysis was carried out by gas chromatography (GC Top 8000, Carlo Erba Instruments, Rodano, Italy) that was equipped with an on-column injection system (injected volume: 1 µL), a WCOT fused silica CP-Sil-8 CB column (50 m × 0.25 mm I.D., film thickness 0.25 µm, Varian Inc., Palo Alto, CA, USA) and a $^{63}$Ni electron capture detector (ECD 80, Carlo Erba Instruments, Rodano, Italy).

The external standards Custom Pesticide Mix (o2si, USA), Custom PCB Calibration Mix (o2si, USA) and Aroclor 1260 (Alltech, Nicholasville, KY, USA) were used for DDT and PCB quantitation. The solution of DDT homologues contained pp'DDT, op'DDT, pp'DDD, op'DDD, op'DDE and pp'DDE, while the analysed PCB congeners were: PCB 18, 28 + 31, 44, 52, 101, 118, 149, 138, 153, 170, 180, 194, and 209. LOD for zooplankton is 0.1 ng $g^{-1}$ dry weight for all compounds.

Routinely, standards reference materials SRM NIST-1947 "Lake Michigan Fish Tissue" and NIST-1946, "Lake Superior Fish Tissue" were analysed in triplicate to test good laboratory practices, respectively, for DDT homologues and PCB residues. The percentage recoveries of DDT were between 106.2 ± 3% and 107.5 ± 4%, while those for PCB ranged from 91.3% (±1.1%) to 102.2% (±1.6%).

*2.4. Data Analysis*

2.4.1. Statistical Analysis

Statistical analysis was performed by R software (R version 3.5.1). For ANOVA analysis, the significance level was set at *p-value* < 0.05. The data were not normally distributed, so they were log-transformed before the analysis. After every analysis, we checked the distribution of residual, according to R package. Principal Component Analysis (PCA) was chosen to describe the internal structure of the data explaining the variance of contaminant concentrations in the dataset. Analysis was performed while using FactoMineR, and factoextra R-packages.

2.4.2. Spatial Analysis

Geometry of basins was obtained from geographical hydrological portal of ARPA-Lombardy and geoportal geo.admin.ch of Swiss Confederation [24,25]. Available spatial data about anthropic pressures were selected for each basin through ArcGIS software (ArcGIS version 10.3.1). In order to describe the study area, the degree of urbanisation (DEGURBA), which classifies local administrative units in three classes, was used. The classes are: densely populated area or cities/large urban area (class 1), intermediate area or towns and sub-urbs/small urban area (class 2), and thinly populated area or rural area (class 3) [26]. For each basin, the percentage of area that is occupied by each class was estimated. We collected also spatial data for wastewater treatment plants (WWTPs) and their dimensions (population equivalent), populations, municipalities and basin areas [27–30].

## 3. Results and Discussion

*3.1. Concentrations of Organic Contaminants*

Figure 2 shows the concentrations of each contaminant group in the different lakes. The results are expressed as sum of 12 congeners of PFAS (ng g$^{-1}$ ww), sum of the two congeners and four respective metabolites of DDT (ng g$^{-1}$ dw) and sum of 14 congeners of PCB (ng g$^{-1}$ dw) in zooplankton samples. Detailed data regarding contaminants concentrations are reported in Tables S4–S6.

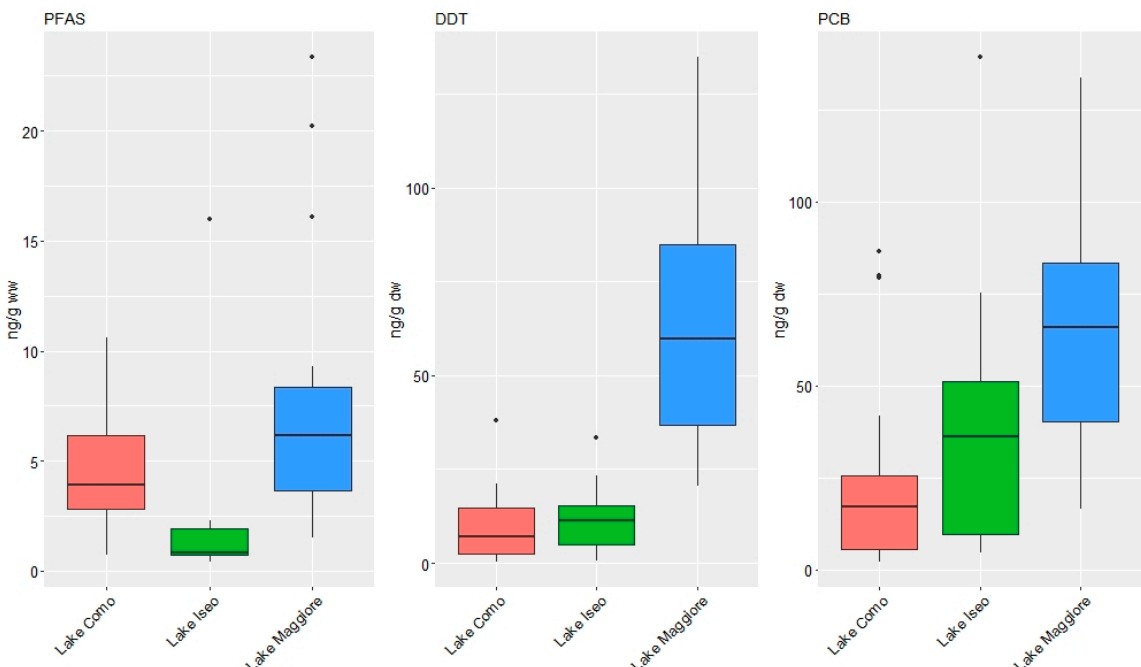

**Figure 2.** Total concentrations of perfluoroalkyl substances (PFAS) (ng g$^{-1}$ ww) and DDT and PCB (ng g$^{-1}$ dw) in zooplankton collected from lakes Como, Iseo, Maggiore.

Differences between lakes were statistically significant (Anova one-way, *p-value* < 0.001 for all compounds; N = 51 for PFAS, N = 72 for OC). In detail, PFAS concentrations in zooplankton from Lake Iseo were lower than those in Lake Como and Maggiore (Tukey test, *p-value* < 0.001), which showed no differences between them; DDT concentrations in Lake Maggiore were significantly higher than in the other lakes (Tukey test *p-value* < 0.001) and PCB levels in zooplankton in Lake Como were lower than in the samples that were collected in Lake Maggiore and Iseo (Tukey test *p-value* < 0.001).

In general, Lake Maggiore showed the highest concentrations for each group of contaminants, with mean values of 7.6 ng g$^{-1}$ ww for PFAS, 65.0 ng g$^{-1}$ dw for DDT, and 65.5 ng g$^{-1}$ dw for PCB.

High levels of DDT and its metabolites in this lake are due to the presence of a point source from a chemical plant located on the River Toce, an important tributary of Lake Maggiore, as already explained in the section "Study Area". The factory produced technical DDT from 1948 to 1996, but the contamination is still present, because these compounds accumulated into the soils around the industrial area [21].

There are not factories that produce PFAS in this area, but lakes are subjected to the effluents of both industrial and urban wastewater treatment plants (WWTPs) and to diffuse pollution from atmospheric deposition. PFAS are not removed in standard treatments of wastewater and enter in water bodies [31]. The basin of Lake Maggiore is characterized by the most extended area, the highest number of inhabitants, and the highest percentage of densely populated area (2%) among the basins of the studied lakes in accord to the European report on degree of urbanization [26] (Table 1). The same ranking of PFAS contamination has been highlighted in fish that were sampled in the same areas [32], and in that work the source of PFAS for Lake Maggiore was hypothesised to be Lake Lugano, which belongs to the Lake Maggiore basin. Lake Iseo, which collects the waters of the smallest basin with the lowest population and number of WWTPs, showed the lowest PFAS concentrations among the studied lakes (mean value: 3.2 ± 5.7 ng g$^{-1}$ ww).

It is more difficult to address the differences in PCB zooplankton concentrations, because the contamination is very old, and no point sources can be identified in the lake basins. In fact, the differences in concentrations cannot be directly related to the basin areas or the inhabitant number. Nonetheless, Lake Iseo has a significantly higher mean concentration of total PCB than Lake Como (40.6 ± 40.1 and 20.9 ± 21.1 ng g$^{-1}$ dw, respectively), and this result could be linked to the great exploitation of hydroelectric power plants in Valcamonica, during the economic development after the second World War, which largely used PCB as dielectric fluids in transformers.

Regarding Italian subalpine lakes, this is the first study of PFAS contamination in zooplankton, while data regarding DDT and PCB are abundant for Lake Maggiore [33], but sporadic for the other lakes. The last determination of DDT and PCB concentrations in zooplankton in Lake Iseo, dating back to 2010, showed that current DDT concentrations are lower, while PCB concentrations are stable [34]. In Lake Como, the comparison with older data showed a decrease in the concentrations of both organo-halogenated compounds [11,35], but there are not enough data to claim a significant decreasing trend.

Concentrations of PFAS in pelagic invertebrates in this study are higher than those that are reported in Baltic Sea [36], where the sums of PFSA and PFCA in zooplankton were only 0.11 ± 0.02 and 0.12 ± 0.01 ng g$^{-1}$ ww, respectively. On the contrary, they are comparable with the concentrations that were measured in the Gironde estuary (France) [37] and in the Arctic Canadian Lakes that are not contaminated by local airport [38].

**Table 1.** Information about the degree of urbanization (DEGURBA), wastewater treatment plants (WWTPs), and population in basins of Lake Maggiore, Como, Iseo.

| Lake Basin | DEGURBA | | WWTPs | | Administrative Data | | |
| --- | --- | --- | --- | --- | --- | --- | --- |
| | Class | Area % | Total WWTPs | Total Population Equivalent | Total Municipalities | Population (2011) | Area (km²) |
| Maggiore | 1 | 2.01 | ND | ND | 207 | 923,861 | 6815.64 |
| | 2 | 21.42 | | | | | |
| | 3 | 76.57 | | | | | |
| Como | 1 | 1.64 | 143 | 759,461 | 191 | 556,769 | 4611.56 |
| | 2 | 18.82 | | | | | |
| | 3 | 79.53 | | | | | |
| Iseo | 1 | 0.00 | 69 | 191,709 | 73 | 191,527 | 1842.48 |
| | 2 | 24.20 | | | | | |
| | 3 | 75.80 | | | | | |

### 3.2. Pattern of Contamination

When considering the composition pattern of PFAS accumulated in zooplankton (Figure 3), PFOS was detected in 96% of the samples and it was the predominant compound in all lakes, reaching the maximum concentration of 18.9 ng g$^{-1}$ ww in Lake Maggiore. It represented 32% of total PFAS concentrations in zooplankton in Lake Iseo, 52% in Lake Como, and 67% in Lake Maggiore. The other two perfluoroalkyl sulfonic acids detected (PFBS, PFHxS) were only determined in significant concentrations in Lake Iseo. Regarding perfluoroalkyl carboxylic acids (PFCA), long-chain compounds (C > 9) predominated in zooplankton, while short-chain compounds (with 6–7 carbon atoms) were only detected in few samples (about 10%) and at lower concentrations (maximum value: 0.9 ng g$^{-1}$ ww). PFOA was detected in about 65% of samples, but it only represented 11.5% of the total PFAS concentration in Lake Iseo, and about 6% in the other two lakes.

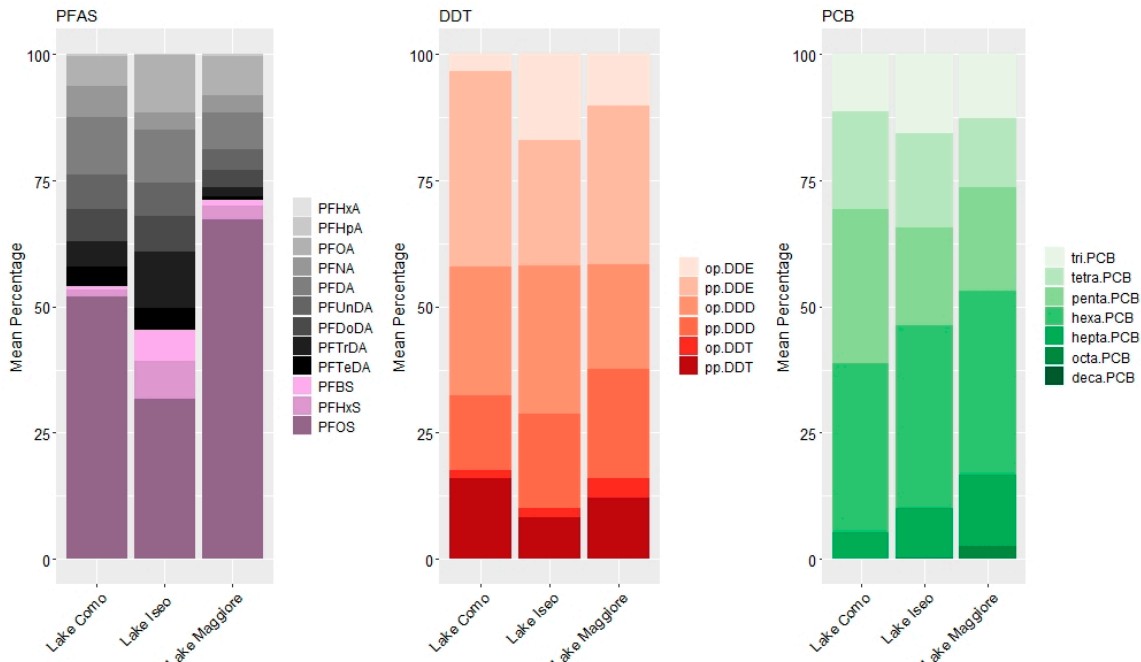

**Figure 3.** Composition pattern of contaminants in zooplankton samples.

Concentrations of PFTrDA (C13) and PFTeDA (C14) were lower than those of other long-chain PFCA, probably because these compounds have higher affinity for particles and sediment is their main sink [39].

We analysed the whole dataset of individual congeners of PFAS by a Principal Component Analysis (PCA) (Figure 4). Loading plot on the first two components, which globally explain 54% of the total variance, helps to identify common behaviour among the individual PFAS congeners (Figure 3). Three different groups are gathered in the loading plot: PFOA, PFBS, and PFHxS compose the first, which is maximum on the second component and orthogonal to the first one. PFOA shows $K_{ow}$ similar to PFHxS [40], and this group of compounds was higher in the samples of Lake Iseo than in the other lakes, which suggests a specific contamination source for this lake. The second group is formed by PFTrDA (C13) and PFTeDA (C14) and it is orthogonal to the first component and parallel to the second one, but in the negative direction. The third group showed PFOS (C8) laying in the same direction of the other long-chain PFCA (8 < C < 13); it is rather orthogonal to the other two groups and it includes the most bioaccumulable and biomagnificable PFAS congeners. In fact, PFBS and PFHxS are the most soluble congeners and PFTrDA (C13) and PFTeDA (C14) are not readily bioavailable because of their molecular size [41,42]. The coefficients in the second eigenvector are correlated with $K_{ow}$ of PFAS substances [40], except for PFOA, which is uncorrelated (Figure S2), suggesting that lipophilicity cannot be used to model bioaccumulation of PFOA in zooplankton.

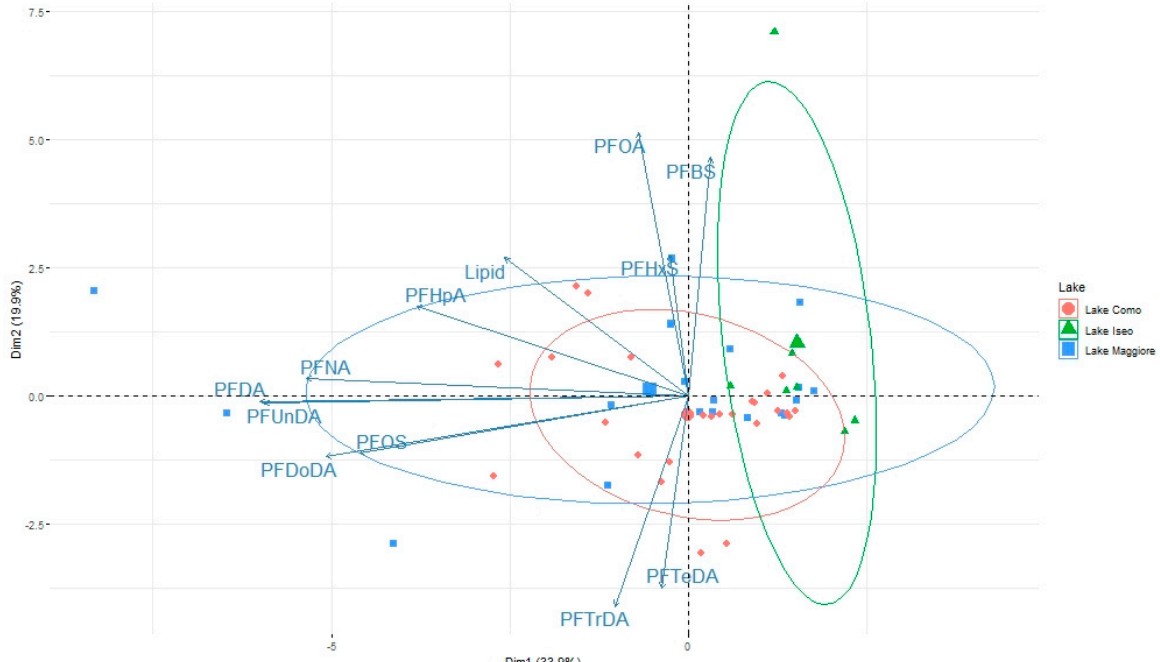

**Figure 4.** Principal Component Analysis (PCA) biplot characterizing individual PFAS in zooplankton samples. Studied variables are mapped with arrows, sample shape is showed in legend. Percentages in brackets refer to the proportion of variance explained by the different axes.

Looking at the bidimensional score plot, lakes Como and Maggiore samples cannot be distinguished, while the Lake Iseo data are better described by the second component where PFOA, PFBS, and PFHxS loadings predominate.

In the group of DDT compounds, metabolites of DDT and their isomers were predominant over the parental compounds (op' DDT and pp' DDT). Technical DDT products generally contained about 75% of pp' DDT, 15% op' DDT, and other compounds in very small amounts. DDT isomers are known to degrade into DDE and DDD under aerobic and anaerobic conditions. Therefore, the increase of the percentage of DDE and/or DDD and a > 1 ratio DDE/DDT indicated that there are no recent inputs to the environment [43]. DDE represented more than 40% of the total concentrations in all lakes and its ratios with DDT were 2.4, 5.3, and 2.3 for Lakes Maggiore, Iseo, and Como, respectively, suggesting that the contamination is old and no recent inputs of parental compound occurred (Figure 3). pp' DDE was

the main compound detected in zooplankton and it was measured in all samples with concentrations that ranged from 0.3 ng g$^{-1}$ dw in Lake Como to 38.3 ng g$^{-1}$ dw in Lake Maggiore (Table S5).

PCB 153 was the congener with the highest frequency of detection (>94%), followed by PCB 101 (91.5%), PCB 44, PCB 180, and PCB 138 (all up to 70% of total samples). In Lake Maggiore, PCB 153 was the congener with the highest concentrations (11.0 ± 8 ng g$^{-1}$ dw), while PCB 149 prevailed in Lake Como (6.6 ± 12 ng g$^{-1}$ dw) and PCB 52 in Lake Iseo (11.0 ±18 ng g$^{-1}$ dw). If we grouped PCB congeners in seven classes based on their number of chlorine atoms, concentrations raised with the increase of number of chlorine atoms until the hexachlorobiphenyl (hexa-CB) group, and then tended to decrease (Figure 3). Accordingly, the prevalent group was hexa-CB, which constituted 35.4% of total PCB concentration, reached a maximum of 60.5 ng g$^{-1}$ dw in Lake Maggiore. The pattern of PCB congeners probably reflected the Aroclor mixtures (Aroclor 1256 e 1260) most used in the past in Italy [44].

While examining loading plot in the PCA of PCB and DDT compounds, gathered in isomer groups (Figure 5), we can see that the coefficients of PCB and DDT isomer groups in the second component are significantly correlated with K$_{ow}$ (Figure S2), except for octachlorobiphenyl (octa-CB). The peculiar octa-CB behaviour cannot be easily explained, but it could be related to the fact that octa-CBs were only determined in Lake Maggiore. As in the case of PFAS, the second component is related to the contaminant lipophilicity and explains 12% of the total variance. Nevertheless, it should be noted that the slope of the correlation between coefficients in the second eigenvector and K$_{ow}$ of PCB and DDT is five-times higher than that interpolated for PFAS (Figure S2).

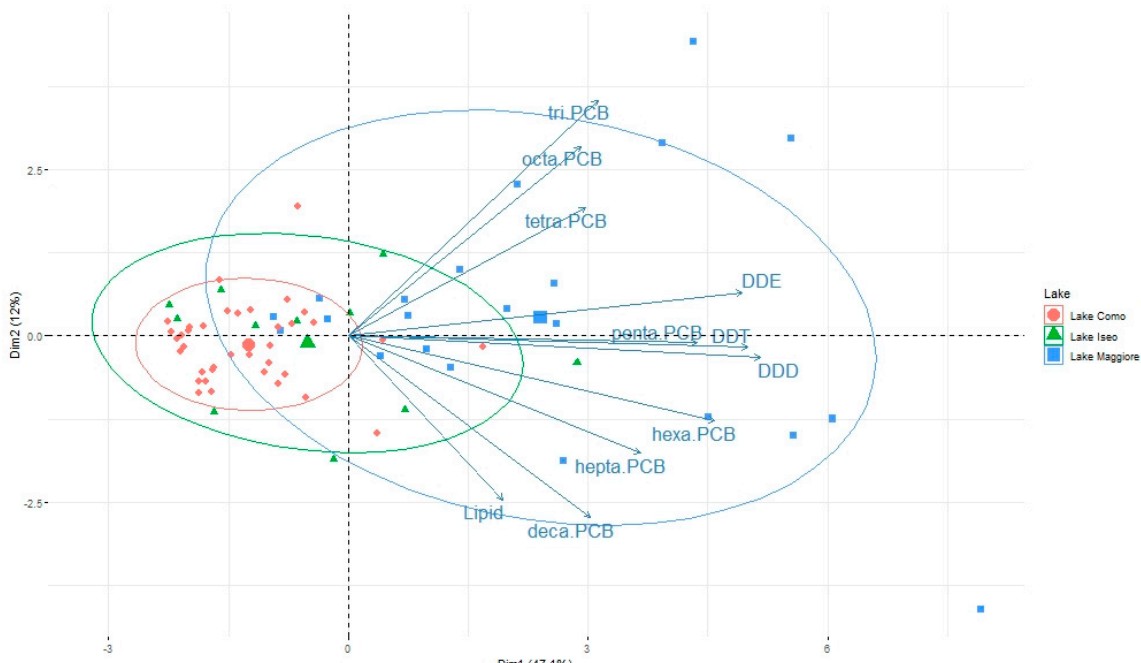

**Figure 5.** PCA biplot characterizing isomer groups of DDT and PCB in zooplankton samples. Studied variables are mapped with arrows, sample shape is showed in legend. Percentages in brackets refer to the proportion of variance explained by the different axes.

The score plot shows that data from different lakes cannot be distinguished, but Lake Maggiore has the highest variability, while the lowest one is shown by Lake Como, as also evident in Figure 2.

### 3.3. Role of Zooplankton Size and Seasonality on Contaminant Levels

Data that were collected in this study allowed for in-depth insight of the role of zooplankton ecology in the contaminant accumulation. Zooplankton has been sampled in different size fractions in order to separate species that are characterized by different trophic levels. Details on size fractions

collected in the different lakes can be found in Table S2 and Table S7 reports data on biomass and taxa composition.

The smallest and the intermediate fractions (≥200 and ≥450 µm) included all crustacean species living in the lakes but had different total biomass, because, in the former, we could collect also the smallest and youngest specimens, having a more complete picture of the zooplankton community. The greatest size fraction (≥850 µm) mainly contained the biggest individuals of Cladocera (generally Daphnia for primary consumers and predators).

We only analysed zooplankton data from Lakes Como and Maggiore, because, for Lake Iseo, there were enough data for the lowest size fraction (200 µm), but the total sampled biomass for the other two fractions was insufficient to complete all the chemical analyses. PFAS data have been analysed as a whole dataset. Since we have shown that there are no statistically significant differences between Lake Como and Lake Maggiore for PFAS data (Figure 2), while for DDT and PCB, the datasets have been separately analysed for each lake (Figure S1).

No significant differences were observed between zooplankton size fractions for all contaminants (Figure S1). According to a biomagnification hypothesis, the biggest fraction, which contains more predators than filter-feeder or herbivores crustaceans, should be the preferred fraction for contaminant accumulation. On the contrary, our results showed that the biggest fraction had no statistically significant differences with the others, and the 850 µm-fraction was clearly less contaminated than the 450 µm-one for DDT and PCB in Lake Maggiore. Piscia et al. [45] suggested that in the smaller fractions there were more copepods, richer in lipids than cladoceran species, and therefore more available to bioaccumulate organic contaminants. Principal Component Analysis of taxonomic compositions and contaminant concentrations, expressed as total concentrations of each chemical family, (Figure 6) showed that chemical concentrations were orthogonal to (i.e., independent from) the taxa of planktonic organisms, but inversely correlated with the total zooplankton biomass and temperature. The score plot showed that the colder seasons (autumn and winter) positively correlated with all of the contaminant concentrations in zooplankton.

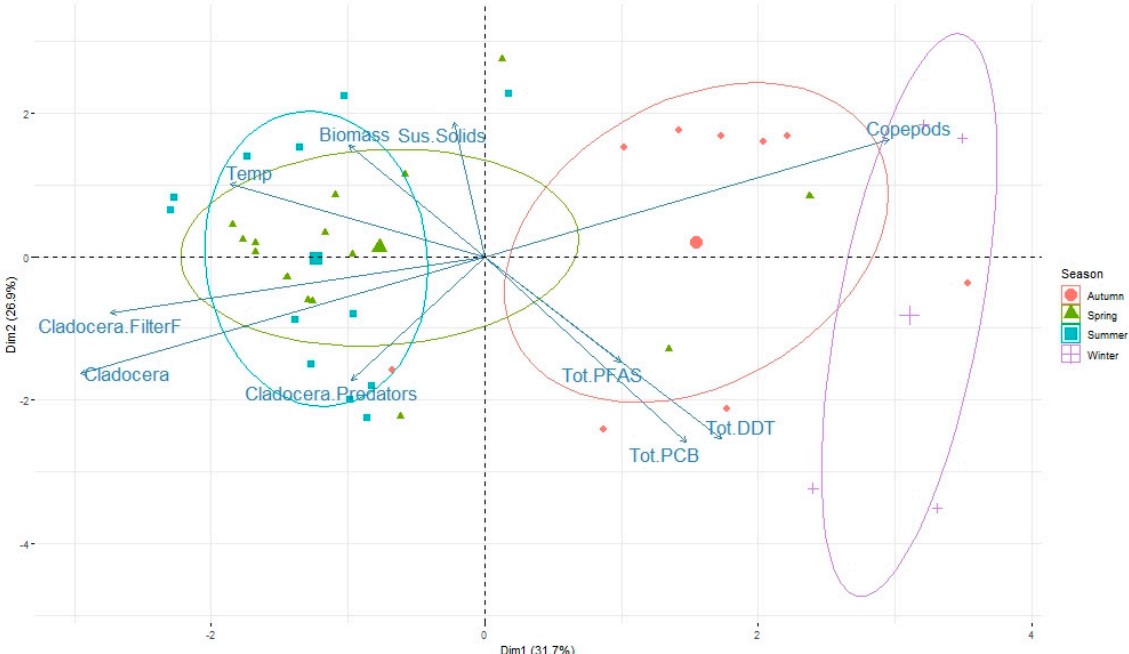

**Figure 6.** PCA biplot characterizing contaminants and biomass of zooplankton samples. Variables are mapped with arrows; sample shape are showed in legend. Percentages in brackets refer to the proportion of variance explained by the different axes.

This result is partially confirmed by comparing concentrations in the different seasons (Figure 7), which shows a similar qualitative trend for all compounds: the concentrations were higher in colder months than in spring and summer, with a characteristic U-shape from winter to autumn. For PFAS, these differences were not statistically significant, while winter DDT concentrations in Lake Como were significantly higher than spring ones ($p$-value < 0.05, Anova and Tukey tests), and, in Lake Maggiore, there were significant differences between winter and both warmer seasons and between autumn and summer. PCB followed the same trend as DDT and both lakes showed significant differences between seasons: in Lake Como ($p$-value < 0.01), there were significant differences between winter and both warmer seasons and between autumn and spring; in Lake Maggiore ($p$-value < 0.05) there were significant differences between summer and colder seasons. No interaction between the considered variables (size and seasons) was evidenced by two-way-Anova test.

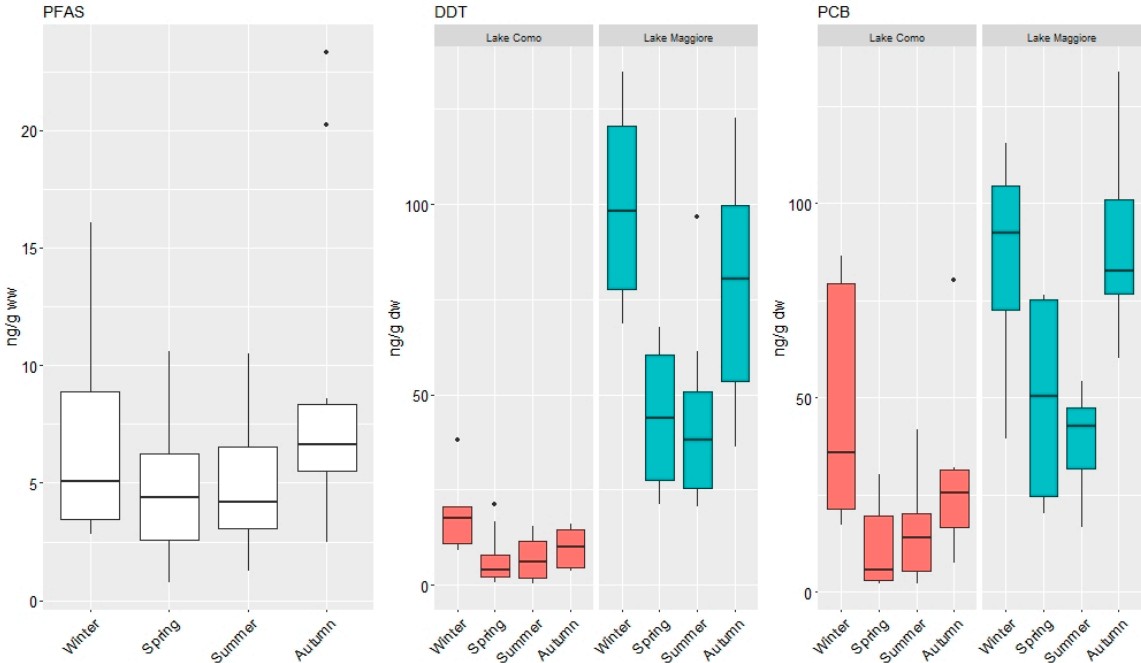

**Figure 7.** Contaminant trend between seasons.

The characteristic U-shaped trend of DDT and PCB concentrations in Lake Maggiore was observed since the beginning of the monitoring activities and it did not vary between years [33]. The inverse relationship between concentrations in zooplankton and zooplankton biomass might be associated with the shift in diet of zooplanktonic specimens because of the different availability of nutrient along the year. Changes in resource availability and environmental conditions (the decrease of food availability or the increase of metabolic costs) can lead to changes in trophic interactions [46]. For example, δ15N‰ of all zooplanktonic species changed along the years, increasing in the cold seasons, as shown in [35]. During spring and summer, phytoplankton is easily available and filter feeders rely on this food source, while, during autumn and winter, they need to eat also bacteria, protozoa, or organic particles to obtain enough energy to live. Additionally, Campbell et al. [47] observed that organisms, which live in cold water from glaciers in an unproductive environment and low nutrients, often become richer in lipid and OC content, indicating that nutrient limitation at the base of the food web can affect the uptake of contaminants at higher trophic levels.

The differences of concentrations throughout the year might be also explained by "the biomass dilution effect", as proposed by Taylor et al. [6], who observed that DDT and PCB concentrations varied across lakes according to an inverse relationship with their planktonic biomass. The same effect, as observed for polycyclic aromatic hydrocarbons in plankton of the Mediterranean and Black Seas,

was explained by a reduction of water concentrations by adsorption on dissolved organic matter and suspended sediments that peak during summer algal bloom [48].

In Italian lakes, which were studied in the present work, the seasonal trend was much stronger for the chlorinated compounds than for PFAS. Variations in PFAS were quite limited, as in the Gironde estuary, where PFAS only varied up to a factor of 2.5× for zooplankton and 2.3× for shrimps in different seasons [49].

## 4. Conclusions

The biannual campaign of monitoring of persistent organic compounds (PCB, DDT, and PFAS) in zooplankton of the Italian subalpine lakes allowed for inferring some conclusions on the relationships among zooplankton ecology, physico-chemical characteristics of the compounds, and bioaccumulation. It was evident that the contaminant concentrations depend on seasonality more than on size, trophic levels, taxa composition, and feeding behaviour of zooplankton. This evidence might indicate that the contaminants are mainly accumulated from water, with a minor contribution from the diet. The good correlation between log $K_{ow}$ and eigenvector coefficients in Principal Component Analysis (Figure S2) showed that a significant driver for the accumulation of most of the studied contaminants is their lipophilicity, except for PFOA and octa-CB.

Analysis of zooplankton, as bulk, could be considered a practical alternative for monitoring purposes using a size mesh that collects more biomass (such as e.g., 200 or 450 μm), both for the description of community composition and for analytical determinations, since the determination of the studied compounds in lake water is often difficult due to their low concentrations and the need for high volume concentrations. Moreover, we suggest sampling during winter or late autumn, when the concentrations are higher, even if the collection of sufficient biomass could require more catches.

**Supplementary Materials:** The following are available online at http://www.mdpi.com/2073-4441/11/9/1901/s1, Table S1: Main morphology characteristics of deep subalpine lakes; Table S2: Monitoring Plan; Table S3: List of PFAS compounds targeted in the present study, corresponding internal standards (ISs) and LC/MS/MS parameters for all target analytes and internal standards; Table S4: PFAS concentrations in zooplankton samples (ng g$^{-1}$ ww); Table S5: DDT concentrations in zooplankton samples (ng g$^{-1}$ dw). LODs are 0.1 ng g$^{-1}$ dw; Table S6: PCB concentrations in zooplankton samples (ng g$^{-1}$ dw). LODs are 0.1 ng g$^{-1}$ dw; Table S7: Taxa composition of zooplankton community in Lake Como and Maggiore; Figura S1: Contaminant trend in different zooplankton size; Figura S2: Correlation between coefficients of 2nd eigenvector in PCA and $K_{OW}$.

**Author Contributions:** Experimental plan, M.Mz. and R.B.; sampling and chemical analysis M.Mz., S.Pa., R.P., M.P., S.V.; data analysis M.Mz, S.Pa, S.Po.; writing—original draft preparation, M.Mz, S.Pa., S.Po.; Review, M.Mz, S.Pa., S.Po., M.Mn., R.B. and S.V.

**Funding:** Zooplankton investigation in Lake Maggiore was funded by CIPAIS (Commissione Internazionale per la Protezione delle Acque Italo-Svizzere).

**Acknowledgments:** Authors thanks Marianna Rusconi for her kind help in Lake Como sampling.

**Conflicts of Interest:** The authors declare no conflict of interest

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
