# Peer review of "Organic Contaminants in Zooplankton of Italian Subalpine Lakes: Patterns of Distribution and Seasonal Variations"

_water, doi:10.3390/w11091901_

Round 1

Reviewer 1 Report

Please see the review report in attached PDF file.

Author Response

ANSWER to REVIEWER1

General Comments

The present manuscript, water-563082, investigated the spatial variability of PFAS, DDT, and PCB contaminants in zooplankton over more than 2 years, and the results were statistically analyzed. PFAS are fluorinated emerging contaminants of high current concern in Europe and elsewhere, while DDT and PCB are legacy halogenated pollutants. There are not many reports on the occurrence of PFAS in low trophic level organisms, and very few studies have examined the influence of seasonal variations. This constitutes one of the key novelty aspects of the study. The analytical chemistry procedures seem solid, and the manuscript is easy to follow with a logical structure and few corrections noted. The suggestion would be to accept the manuscript for publication pending minor revisions.

Thank you for your positive opinion

Section Comments Introduction

Before line 32, | think the authors should add an opening paragraph in introduction regarding key halogenated contaminants targeted in this study. Basically, the authors want to state that these are anthropogenic contaminants, with persistent, bioaccumulative, and toxic properties. What is the current state of these contaminants in the European Union? Most of them are supposed to be banned but are still found in aquatic ecosystems in Europe and elsewhere. Also include related literature.

We added a sentence and some references in the introduction to maintain the structure of the speech

Line 41. Update the reference format.

We modified the reference

Line 51. “On the contrary researches”. Suggest to replace by “Previous studies”.

We modified the sentence

Line 53. Edit for grammar: “has been analyzed”.

We corrected the sentence

Materials and methods

Line 67. “constitutes”. Need to check subject-verb agreement.

We corrected the word

Line 69. Missing word? (“because they have”)

We corrected the sentence

Line 72. “satisfies”. Need to check subject-verb agreement.

We modified the word

Line 72. | think it would be interesting to give some examples of municipalities/cities that use the surface water from these lakes to produce drinking (tap) water.

We added the examples

Line 84. until the 1990s.

We corrected the sentence

What software was used to draw Fig.1? Perhaps add a note for the source of the base maps.

We added the information in the caption

Line 90. Remove the parenthetical statement “(to grasp main changes in biomass)”

We removed the sentence

Line 105. 5g of dry weight or 5g of wet weight? Please clarify.

We added the information

Line 114. Replace “obtain” by “construct”

We changed the word

Line 117-119. Revise the sentence for grammar and syntax.

We rewrote the sentence

Lines 140-144. Were PFAS also analyzed in these SRM fish matrixes? It would have been interesting to see the data.

No, we have not used certified material for PFAS in biological matrix. We use recovery data from real matrices, as detailed in Mazzoni et al., 2016

Line 152. Please edit to: “Analysis was performed using FactoMineR and factoextra R-packages.” We corrected the sentence

Line 160. “thinly populated”. Consider editing to: “sparsely populated”.

“Thinly populated” is the definition used in the European Commission Regional Working Paper 2014 that we cited. We prefer to adopt the language used in the document.

Line 161. Word order. We also collected.

We changed the order

Results and discussion

Line 188. The basin of...

We modified the sentence

Line 196. “It's”. Revise from grammar (avoid contractions).

We modified the sentence

Lines 201-202. 1960s and 1970s.

We modified the sentence

Line 208. we do not have.

We modified the sentence

Lines 214-223. Provide some comparative discussion with PFAS levels and profiles in aquatic invertebrates from literature. Some recent studies that could be considered:

https://pubs.acs.org/doi/abs/10.1021/es5048649

https://www.sciencedirect.com/science/article/pii/S0048969719325161

https://pubs.acs.org/doi/abs/10.1021/acs.est.6b01197

https://pubs.acs.org/doi/abs/10.1021/acs.est.7b02399

We add a comparative discussion citing the suggested references

Lines 300-301. Was the statistical test performed for summed PFAS, or individual PFAS? Please clarify.

We specified in the text that “we analysed the whole dataset of individual congeners of PFAS by a Principal Component Analysis (PCA) (Fig. 4)”

Line 370. Add a reference? Why steady state conditions would be achieved in autumn/winter seasons?

We rewrote the text, because we recognize that many sentences are not supported enough by data, especially about the reaching of the steady state for bioconcentration in colder months. We eliminated I also from the conclusions. We add different hypothesis to explain this experimental evidence.

 Supporting Information

Table S2. Number of samples per size. Number of samples per season.

We corrected the sentence

Can you add the details on the PFAS chromatographic method (LC gradient elution program, injection volume, etc.).

The details are in Mazzoni, M.; Polesello, S.; Rusconi, M.; Valsecchi, S. Liquid chromatography mass spectrometry determination of perfluoroalkyl acids in environmental solid extracts after phospholipid removal and on-line turbulent flow chromatography purification. J. Chromatogr. A 2016, 1453, 62–70, doi:10.1016/j.chroma.2016.05.047

If available, can the authors add a table with PFAS spike recovery values to biota matrix? Also, suggest to add a table with determined concentrations in NIST SRM samples.

As already said, we were not able to use SRM for PFAS in biological matrix, but recovery tests are reported in cited work Mazzoni et al., 2016.

Reviewer 2 Report

The reported study is of great interest and the scientific methods utilized appear sound.  

A few major concerns that need to be addressed prior to publication include:

1) large number of writing errors and some of the organization is weak.

2) the abstract does not offer statements that preface the study.

3) it is important to analyze the current persistent contaminants separate from the legacy contaminants, since one is likely still contributing to the contaminant load in the lakes, while the others are like not. While the study is focused on the contribution by zooplankton, the authors make statements such as the last one in the abstract - "Principal Component Analysis showed that zooplankton bioconcentrates (contaminants) from water, without differences in feeding behaviour." It can be argued that it is important to know the concentration of the contaminants in the water as well as the zooplankton.

4) can the authors quantify the number of people who rely on the lake water?

5) can the authors offer more details on the sampling and the purpose of the 20 m depth for collection?

6) overall, I am having difficult accepting the overall conclusions of the study.   

Author Response

ANSWER TO REVIEWER2

Extensive editing of English language and style required

We have carried out a complete revision of the text, both grammatical and stylistical. We rephrased many sentences, as you can find in the trackchange version.

Comments and Suggestions for Authors

The reported study is of great interest and the scientific methods utilized appear sound. 

Thank you for your positive opinion

A few major concerns that need to be addressed prior to publication include:

1) large number of writing errors and some of the organization is weak.

We corrected them

2) the abstract does not offer statements that preface the study.

We added a sentence at the beginning.

3) it is important to analyze the current persistent contaminants separate from the legacy contaminants, since one is likely still contributing to the contaminant load in the lakes, while the others are like not. While the study is focused on the contribution by zooplankton, the authors make statements such as the last one in the abstract - "Principal Component Analysis showed that zooplankton bioconcentrates (contaminants) from water, without differences in feeding behaviour." It can be argued that it is important to know the concentration of the contaminants in the water as well as the zooplankton.

We are not able to sample and analyse waters. In sporadic analyses we found most of the compounds around or below the detection limits. Since the dataset is largely uncomplete we omit to include in the paper. But we agree with the reviewer that we cannot speak about bioconcentration without data on water. On the consequence we modified the text in the “abstract”, in the “results and discussion” and in the “conclusions” sections and we eliminated the sentences related to bioconcentration.

4) can the authors quantify the number of people who rely on the lake water?

We wrote the number of inhabitants in Table 1. Instead, we can’t know the number of people that use the water lake as drinking water, we know the province that we added on the text

5) can the authors offer more details on the sampling and the purpose of the 20 m depth for collection? We added some information

6) overall, I am having difficult accepting the overall conclusions of the study.

We rewrote the conclusions, because we recognize that many sentences are not supported enough by data, especially about the bioconcentration and the reaching of the steady state for bioconcentration in colder months.
